# Enhanced Microbial Oxidation–Neutralization Treatment of Acid Mine Drainage Rich in Ferrous Ions (Fe^2+^)

**DOI:** 10.3390/ijerph19116543

**Published:** 2022-05-27

**Authors:** Wenjie He, Haibo Li, Yin Xu, Feng Zhong, Hao Dong, Min Wang

**Affiliations:** Faculty of Resources and Environmental Science, Hubei University, Wuhan 430062, China; hewenjie@stu.hubu.edu.cn (W.H.); lhb@hubu.edu.cn (H.L.); yxu@hubu.edu.cn (Y.X.); zhongfeng@stu.hubu.edu.cn (F.Z.); donghoah@stu.hubu.edu.cn (H.D.)

**Keywords:** ferrous ions, acid mine drainage, *A.* *ferrooxidans*, oxidation, neutralization

## Abstract

In this work, a method of enhanced packed-bed microbial oxidation–neutralization has been employed to treat Fe^2+^-rich acid mine drainage. The method features the use of a large number of immobile *Acidithiobacillus ferrooxidans* (*A. ferrooxidans*) in a bioreactor to promote the oxidation of Fe^2+^ to Fe^3+^. Results show that when the influent Fe^2+^ concentration is about 900 mg/L and the Fe^2+^ oxidation efficiency tends to 100%, the maximum oxidation rate of Fe^2+^ in the bio-ceramsite, bio-volcanic stone, and bio-activated carbon packed columns are 301 mg/(L·h), 234 mg/(L·h), and 139 mg/(L·h), respectively. Compared with the direct neutralization method, the enhanced microbial oxidation–neutralization method has several advantages. Firstly, it oxidizes Fe^2+^ to Fe^3+^, directly neutralizing the acid mine drainage at low pH and reducing the consumption of neutralizer. Secondly, more economical CaCO_3_ can be used as neutralizer. Thirdly, it produces precipitates with high solid content (5.50%), good settling performance (SV_30_ = 4%), and small volume, and the capillary suction time (CST) is 8.9 s, which is easy to dehydrate.

## 1. Introduction

Acid mine drainage is one of the most serious environmental problems facing the global mining industry [1,2]. It is formed when sulfide minerals are exposed to water, air, microorganisms (*Thiobacillus thiooxidans* and *Thiobacillus ferrooxidans*), etc., and is characterized by high acidity and heavy metal content [3,4,5,6]. At present, researchers around the world have explored many methods and materials to treat acid mine drainage, some of which have achieved good results. Among them, Canadian scholars have used microbial fuel cells and electrocoagulation, chlorella, and other methods to treat acid mine drainage. The removal efficiency of Fe^3+^ can reach 94%, and the removal of Cu, Co, and other heavy metals has also achieved remarkable results [7,8]. Scientists in South Africa tried to use synthetic struvite to remove Al^3+^, Fe^3+^, and Mn^2+^ in acid mine drainage, and the removal efficiency could reach 98.99% [9]. Researchers in Mexico raised the pH of acid mine drainage from 2.5 to 4.12 using dual-chamber microbial fuel cells, removing 84% of Cu and 71% of Fe [10]. In addition, some researchers found that 73% of sulfates and 99% of heavy metals in acid mine drainage (gold mine) were removed by sequential hydrotalcite precipitation and sulfate-reducing bacteria [11]. A novel biochar-clamshell composite had good adsorption and economical practicability for Al in acid mine drainage [12]. The crab shell was used as a carrier for microorganisms to treat acid mine drainage, which could remove 90% of Al^3+^ and Fe^3+^, and 50% of Mn and Zn [13]. In China, some researchers used a multi-stage constructed wetland enhancing effluent pH (from 2.5–8.1) and removing dissolved metals (89.4% of Mn, and 99% of Fe, Zn, Cd, and Cu) [14,15,16,17]. However, most of the mentioned methods and results cannot be widely adopted in engineering treatments, owing to governance cost. Therefore, seeking an efficient processing technology for industrial treatment of ferrous ion-rich acid mine drainage is still a big challenge.

The composition of acid mine drainage varies with the type and content of the mine, and the containing pollutants are also different. Usually, acid mine drainage rich in Fe^2+^ was very common, such as in coal mines (Longshan, China, Fe^2+^ = 200 mg/L) [18] or uranium mines (Xinjiang, China) (Fe^2+^ = 323 mg/L) [19]. The traditional treatment method for this type of acid mine drainage was to increase the pH and use CaO as a neutralizer to completely settle Fe^2+^, resulting in a large and difficult-to-dewater sludge with high disposal costs [20,21]. Compared with Fe^2+^, Fe^3+^ can be completely precipitated at a lower pH. Thus, using *Acidithiobacillus ferrooxidans* (*A. ferrooxidans*) to rapidly oxidize Fe^2+^ to Fe^3+^ has great application values [22]. However, because the free state of *A.*
*ferrooxidans* can be easily lost by hydraulic shock, immobilization is usually used to increase its density, thereby accelerating the oxidation efficiency of Fe^2+^ [23].

In this work, we focus on the effect of an enhanced microbial oxidation–neutralization method in the treatment of Fe^2+^-rich acid mine drainage. Firstly, in the oxidation unit, bio-ceramsite, bio-volcanic stone, and bio-activated carbon were used as the immobile carrier of *A. ferrooxidans*, respectively, to increase the density of the bacteria, aiming at obtaining a stable and high Fe^2+^ oxidation efficiency. Secondly, the simulated acid mine drainage oxidized by *A. ferrooxidans* was subject to a neutralization reaction, and the amount of neutralizer (lime or limestone) requiring for complete neutralization was determined. Thirdly, the pH of effluent before and after the neutralization reaction, the content of soluble Fe ions, and the resulting mud the sedimentation performance and sludge dewatering performance were also explored. Finally, the raw water that had not passed through the biological oxidation unit was used as a control for benefit analysis.

## 2. Materials and Methods

### 2.1. Simulated Acid Mine Drainage

The analog acid mine drainage was prepared with reference to 9K medium. The liquid culture medium contained the following basalsalts: (NH_4_)_2_SO_4_ 3.0 g/L, KCl 0.10 g/L, K_2_HPO_4_ 0.50 g/L, Ca(NO_3_)_2_·4H_2_O 0.01 g/L, MgSO_4_·7H_2_O 0.50 g/L. All the chemicals with analytical grade were purchased from Sinopharm Chemical Reagent Co., Ltd. (Shanghai, China). The initial Fe^2+^ concentration (added in the form of FeSO_4_·7H_2_O) was set to 900 mg/L, 600 mg/L, and 300 mg/L, respectively, corresponding to three phases of reaction. Unless otherwise specified, the initial pH was adjusted to 1.8 with dilute H_2_SO_4_.

### 2.2. Microorganisms and Their Fixation

The material for the immobilized carrier was bio-activated carbon (granular, particle size of about 0.1 cm), bio-ceramsite (spherical shape, particle size of 0.5–1.5 cm), and bio-volcanic stone (porous basalt, particle size of 0.5–1.0 cm).

Before use, the above three carriers were soaked in 1 mol/L H_2_SO_4_ for 24 h, rinsed repeatedly, placed in a bucket, and then the *A.*
*ferrooxidans* common bacterial liquid and 9K medium (Fe^2+^ = 8960 mg/L, pH = 1.8) were added. After the complete oxidization of Fe^2+^ in solution under aeration, the fresh liquid culture medium was updated, and then the medium was inoculated with a small amount of bacterial solution, and aeration of the culture was performed again. After the second oxidation was complete, we replaced the fresh medium without inoculating the bacterial solution, and used the bacteria adsorbed on the carrier as the inoculum for the reaction. After repeated cultivation several times, the Fe^2+^ oxidation rate was no longer significantly increased, implying that *A. ferrooxidans* has been successfully immobilized on the carriers, and the carriers can be transferred to the packed bed bioreactor for subsequent reaction.

### 2.3. Packed-Bed Bioreactor

This experiment involved the use of a gas–liquid co-current operation, i.e., the simulated acid mine drainage and air entered from the bottom of the reactor, and then flowed freely upwards in the packing layer, with the air being directly discharged. The oxidized simulated acid mine drainage overflowed from the water outlet and was discharged into a low tank, where it was collected. The reactor was officially in operation for 54 days. Due to the operation in winter, the water temperature was maintained at about 18 °C, and the whole process was divided into three phases. Phase I: from day 1 to day 29, the influent Fe^2+^ concentration was about 900 mg/L; Phase II: from day 30 to day 45, the influent Fe^2+^ concentration was about 600 mg/L; and Phase III: from day 46 to day 54, the influent Fe^2+^ concentration was about 300 mg/L.

The inlet water flow rate was controlled by a valve (medical infusion tube). At the initial stage of the reaction, the flow rate was low and the hydraulic retention time (HRT) was longer. After the Fe^2+^ in the outlet water was stable, the flow rate was increased slowly, to shorten the HRT. The pH, Fe^2+^, and total Fe ions (TFe) of the incoming and outgoing water were dynamically monitored before and after oxidation. The flowchart for the experimental device is shown in Figure 1.

The packed bed in the bioreactor was made of plexiglass, with an inner diameter of 7 cm, and a total of three pieces. Two of the pieces were 47 cm high, with a water outlet every 5 cm from the upper end. The inner diameter of the outlet was 0.5 cm, and there were four water outlets in total. The other piece was 54 cm high, with four outlets every 5 cm from the upper end. At the same time, a sieve plate was added 7 cm away from the lower end, to act as a supporting layer. A rubber stopper was added at the bottom of the plexiglass tube, and then the tube was filled with glass beads and glass wool, in order to prevent the precipitation of pyrite or the carrier from blocking the water inlet pipe or air inlet pipe during actual operation.

After the successful trial operation of the device, the bio-ceramsite, bio-activated carbon, and bio-volcanic stone adsorbed with *A.*
*ferrooxidans* were respectively filled into the three beds, and the device was set for 24 h continuous operation, flowing into the simulated acid mine drainage. The effective volumes of the packed beds of bio-ceramsite, bio-volcanic stone, and bio-active carbon were 1230 mL, 1246 mL, and 1246 mL, respectively, and the void volumes (volume of liquid contained) were 500 mL, 700 mL, and 600 mL, respectively.

### 2.4. Neutralization Reaction

The effluent after oxidation was collected for neutralization reaction. The neutralizer used lime (CaO, analytically pure) and limestone (CaCO_3_, analytically pure). Different amounts of neutralizer were added to a unit volume (150 mL) of water, and the amount of neutralizer used and the variation in the pH, Fe^2+^, and TFe of the effluent after neutralization were recorded. In addition, the sedimentation (sludge velocity (SV_30_)) and dewatering performance (sludge capillary suction time (CST)) of the sludge were simultaneously analyzed after neutralization. Furthermore, the simulated acid mine drainage raw water without undergoing the oxidation reaction was used as a control, and the neutralization reaction was carried out as before, to analyze the relevant indicators.

### 2.5. Determination Method

The pH value was monitored using a pHS-3C model digital pH meter (Shanghai Lei Magnetic Factory). The Fe^2+^ and TFe contents were measured by a colorimetric procedure using 1, 10-phenanthroline as described in standard methods (APHA, 1995). The dried precipitates was characterized by powder X-ray diffraction (XRD) (Rigaku Rotaflex D/max, Japan) by using CuKα radiation (50 kV, 150 mA). The surface morphology of precipitate were observed and photographed by using a scanning Electron Microscopy (SEM) (Hitachi S-3000N, 15 KV). The sludge CST (which refers to the time taken for the water in the sludge to penetrate 1 cm into the Whatman Grade 17 Chr; unit: s) was measured using a CST instrument (British Triton Company, 304M). For the SV_30_, the neutralized mixture was thoroughly shaken and then poured into a 100-mL graduated cylinder, where it was then left to stand for 30 min. The volume of the precipitate was then recorded and the percentage of the volume of the original mixture was calculated.

## 3. Results and Discussion

### 3.1. Solid Characterization

#### 3.1.1. XRD

X-ray diffraction analysis (XRD) is the most effective means to identify mineral species. Since the medium system of the three carriers is the same, one is selected as a representative. As can be seen in Figure 2, the characteristic diffraction peaks of *A. ferrooxidans* oxidation product mostly coincide with that of pyrite. By retrieving the Powder Diffraction File (PDF-4/Minerals+2018) of the International Centre for Diffraction Data (ICDD), these “sharp” diffraction peaks of solid product are assigned to ammoniojarosite (NH_4_Fe_3_(SO_4_)_2_(OH)_6_, JCPDS File No. 26-1014) in the jarosite series.

#### 3.1.2. SEM

The initial pH in the range of 2.5–4.5 is favorable for the formation of Schneider’s minerals, and the pH < 2.5 is favorable for the formation of jarosite [24]. However, pH < 1.5 is harmful to the growth of *A. ferrooxidans*. At pH < 1.0, the activity of *A. ferrooxidans* was completely inhibited [25]. The suspended solids in the effluent of the packed column were analyzed by scanning electron microscopy (SEM). As shown in Figure 3a,b, the product pyrite generated by ferrous ions under the action of *A. ferrooxidans* was mostly micron-sized and formed by agglomeration of many small angular particles. The circled part was *A. ferrooxidans* attached to pyrite (Figure 3c,d).

### 3.2. The Oxidation Effect of Bio-Volcanic Stone on Fe^2+^

Figure 4 depicts the variation of the Fe^2+^ concentration, Fe^2+^ oxidation efficiency ((C_0_ − C_t_)/C_0_ × 100%), Fe^2+^ oxidation rate ((C_0_ − C_t_)/HRT), and HRT in the bio-volcanic stone packed column. On the first day, the influent Fe^2+^ was 849 mg/L and the effluent Fe^2+^ was 0. Although the oxidation efficiency of Fe^2+^ was 100%, the oxidation rate of Fe^2+^ was only 61 mg/(L·h), due to the high HRT (14.00 h). We then slowly increased the inlet water flow rate and increased the hydraulic load. By the 20th day, the HRT had dropped to 5.20 h, and the Fe^2+^ oxidation rate had increased to 147 mg/(L·h). At this time, the effluent Fe^2+^ was 0 and the Fe^2+^ oxidation efficiency was 100%. In the process of adjusting the HRT, the effluent Fe^2+^ fluctuated up and down, but as long as the effluent Fe^2+^ concentration was no greater than 90 mg/L, the Fe^2+^ oxidation efficiency was above 90%.

In the presence of the culture medium, *A. ferrooxidans* needs energy for growth and reproduction, and the generation time of *A. ferrooxidans* is 6.00–10.00 h [26], so when the HRT is less than 6.00 h, the number of free bacteria in the reactor will continue to decrease. From Figure 4, during days 1–20, although the HRT was continuously decreasing, the oxidation rate of Fe^2+^ still showed an upward trend. Since the oxidation of Fe^2+^ during this period mainly came from the action of the fixed *A. ferrooxidans*, the biofilm continued to mature during this period, and the number of bacteria was also increasing.

When the reaction progressed to the 22nd day, the HRT was shortened to 3.50 h and the Fe^2+^ oxidation rate increased to 225 mg/(L·h). At this time, the effluent Fe^2+^ was as high as 200 mg/L, but the Fe^2+^ oxidation efficiency was only 75%. In order to verify the buffering capacity and adaptability of the device under high hydraulic load conditions, the HRT was deliberately controlled at a low level on the 23rd to 26th days, for 2.85, 2.53, 2.38, and 2.30 h, respectively. From Figure 4, it can be seen that although the oxidation rate of Fe^2+^ during this period showed an upward trend, reaching a maximum of 373 mg/(L·h), the oxidation efficiency of Fe^2+^ continued to be around 70%, and the concentration of Fe^2+^ in the effluent was always greater than 200 mg/L. In order to control the concentration of Fe^2+^ in the effluent, the HRT was extended to 3.40 h on the 28th day. At this time, the influent Fe^2+^ was 814 mg/L, the effluent Fe^2+^ was 20 mg/L, the Fe^2+^ oxidation efficiency was 98%, and the Fe^2+^ oxidation rate was 234 mg/(L·h). Summarizing the reaction law during days 1–28, it can be seen that, under the condition of ensuring the Fe^2+^ oxidation efficiency, the 28th day should be the best reaction state for the bio-volcanic stone packed column, when the influent Fe^2+^ is about 900 mg/L. Although further shortening the HRT can increase the Fe^2+^ oxidation rate, when the HRT is less than 3.00 h, the effluent Fe^2+^ concentration is higher, and the reactor no longer has practical application significance.

On the 31st day, the influent Fe^2+^ was adjusted to about 600 mg/L. According to the actual operation of the device, the HRT was controlled at two levels of 7.00 h and 4.67 h. It can be seen from Figure 4 that, during days 32–34, when the HRT was 7.00 h, almost no Fe^2+^ was observed in the effluent and the Fe^2+^ oxidation efficiency was 100%. On the 36th day, the HRT was shortened to 4.67 h, and the effluent Fe^2+^ increased to 134 mg/L. After a period of adaptation, the effluent Fe^2+^ returned to 0 by the 41st day. After the 46th day, the influent Fe^2+^ was adjusted to about 300 mg/L, and the HRT was mainly controlled at two levels of 4.67 h and 2.33 h. From Figure 4, it can be seen that the stability of the system is good, and the Fe^2+^ concentration in the effluent is low.

In summary, when the influent Fe^2+^ drops from 900 mg/L to about 600 mg/L and 300 mg/L, the oxidation rate of Fe^2+^ decreases significantly, and the average value reaches only 100 mg/(L·h). In addition, the reactor does exhibit a certain degree of fluctuation during operation, but after a period of adaptation, the Fe^2+^ concentration in the effluent can still be maintained at a low level, indicating that the bio-volcanic stone packed column has a certain impact resistance.

### 3.3. The Oxidation Effect of Bio-Ceramsite on Fe^2+^

Figure 5 describes the variation of the Fe^2+^ concentration, Fe^2+^ oxidation efficiency, Fe^2+^ oxidation rate, and HRT in the bio-ceramsite packed column. Compared with bio-volcanic stone, the Fe^2+^ oxidation ability of bio-ceramsite is stronger. During days 1–16 of the reaction, the HRT gradually decreased from 10.00 h to 2.05 h, and the oxidation rate of Fe^2+^ showed an upward trend, reaching 400 mg/(L·h) by the 16th day. The oxidation efficiency of Fe^2+^ was also more than 93%, and the Fe^2+^ in the effluent remained below 50 mg/L.

During days 17–23, the effluent Fe^2+^ fluctuated greatly, so we increased the HRT to help the system return to normal. By the 23rd day, the influent Fe^2+^ was 871 mg/L, the HRT was 2.89 h, the effluent Fe^2+^ was 0, the oxidation efficiency of Fe^2+^ was 100%, and the oxidation rate of Fe^2+^ was 301 mg/(L·h).

On the 24th day, when the HRT was shortened to 1.71 h, the Fe^2+^ in the effluent suddenly increased to 450 mg/L, and only 47% of the Fe^2+^ was oxidized. By maintaining a low HRT until the 27th day, the effluent Fe^2+^ remained in a high state, and the Fe^2+^ oxidation efficiency was only about 50%. During days 28–30, we increased the HRT to allow the system to buffer, and the effluent Fe^2+^ was reduced, to a certain extent.

From days 31 to 45, when the influent Fe^2+^ was adjusted to about 600 mg/L, and the HRT was controlled at 5.00 h and 3.30 h, the effluent Fe^2+^ and Fe^2+^ oxidation efficiency were always in a state of great fluctuation. Specifically, when the HRT was 5.00 h (days 31–35), the effluent Fe^2+^ reduced to 0. When the HRT was controlled at 3.30 h, the effluent Fe^2+^ was always above 200 mg/L.

During days 46–54, the influent Fe^2+^ was adjusted to about 300 mg/L. It can be seen from Figure 5 that, when the HRT was shortened to 1.67 h during days 50–54, the Fe^2+^ effluent on the 50th day was 89 mg/L, the Fe^2+^ oxidation efficiency was only 67%, and the Fe^2+^ oxidation rate was 110 mg/(L·h). After the 51st day, the Fe^2+^ oxidation efficiency increased to more than 80%, the effluent Fe^2+^ was controlled at about 50 mg/L, and the Fe^2+^ oxidation rate showed an upward trend, reaching 158 mg/(L·h) on the 54th day. It can be seen that, with the extension of the reaction time, the system gradually becomes stable, the oxidation capacity continues to increase, and the concentration of Fe^2+^ in the effluent becomes lower and lower.

### 3.4. The Oxidation Effect of Bio-Activated Carbon on Fe^2+^

Figure 6 depicts the variation of the Fe^2+^ concentration, Fe^2+^ oxidation efficiency, Fe^2+^ oxidation rate, and HRT in the bio-activated carbon packed column. Compared with bio-volcanic stone and bio-ceramsite, bio-activated carbon has the worst oxidation ability. At the beginning of the reaction (days 1–15), when the influent Fe^2+^ was about 900 mg/L, the oxidation rate of Fe^2+^ was basically lower than 40 mg/(L·h). In order to ensure that the concentration of the effluent Fe^2+^ is low, the HRT should be more than 15.00 h.

Bio-activated carbon reacts well in the process of aeration and film formation, but the oxidation capacity of bio-activated carbon in the packed column is low. The main reason for this is that the supply of O_2_ in the reactor is uneven. Differing from bio-ceramsite and bio-volcanic stone, which have large particles and many pores, the particle size of bio-activated carbon is smaller. After being loaded into the reactor, some areas of the carrier may be relatively dense, and the amount of aeration is not enough to make the carrier fluidized. Therefore, when the air passes through the carrier, it mainly passes through the areas with large gaps in the reactor.

From the 16th day onwards, we increased the aeration rate of the bio-activated carbon packed column. It can be seen from Figure 6 that the oxidation rate of Fe^2+^ showed an upward trend. During days 16–30, with the adjustment of the HRT, the effluent Fe^2+^ showed large fluctuations, indicating that, when the HRT was 6.00 h and above, the reactor operation effect was better.

From days 31–45, the influent Fe^2+^ was about 600 mg/L. During days 31–35, when the HRT was 6.00 h, the Fe^2+^ oxidation rate gradually increased to 95 mg/(L·h), and the effluent Fe^2+^ was 0. During days 36–44, when the HRT was 4.00 h, the oxidation efficiency of Fe^2+^ and Fe^2+^ in the effluent fluctuated greatly, but under the premise of ensuring the oxidation efficiency of Fe^2+^, the oxidation rate of Fe^2+^ was maintained at about 110 mg/(L·h).

On days 46–54, the influent Fe^2+^ was adjusted to about 300 mg/L. When the HRT was 4.00 h and 3.00 h, the effluent Fe^2+^ was 0. When the HRT continued to drop to 2.00 h, the effluent Fe^2+^ suddenly rose to 75 mg/L (the 50th day), but as time passed by, the system became stable. When the effluent Fe^2+^ was controlled below 40 mg/L, the Fe^2+^ oxidation efficiency was about 90%, and the Fe^2+^ oxidation rate gradually rose to 136 mg/(L·h).

### 3.5. The Change of the Fe^2+^ Oxidation Rate in the Different Packed Columns

Figure 7 depicts the variation of the Fe^2+^ oxidation rate in the bio-ceramsite, bio-volcanic stone, and bio-active carbon packed columns under different influent Fe^2+^ concentrations and HRT conditions. During days 1–30 (Phase I), when the influent Fe^2+^ was about 900 mg/L, the Fe^2+^ oxidation capacity of the three carriers was in the order of bio-ceramsite > bio-volcanic stone > bio-activated carbon. With the operation of the system and the shortening of the HRT, the rate of the Fe^2+^ oxidation showed an overall upward trend. On the premise of stable operation of the system, the best state of the above three packed columns was selected in Phase I (see Table 1).

Bio-ceramsite, bio-volcanic stone, and bio-activated carbon all have the characteristics of a rough surface, developed micropores, and a large specific surface area, so that they are particularly suitable for the growth and reproduction of microorganisms on their surface. Therefore, they are widely used as immobilized carriers in water treatment, to increase the density of the microorganisms and reduce bacterial loss [27,28,29,30,31].

In addition, it can be clearly seen from Figure 7 that the oxidation rate of Fe^2+^ in each reactor is positively correlated with the concentration of Fe^2+^ in the influent, to a certain extent. When the influent Fe^2+^ dropped from 900 mg/L to about 600 mg/L and 300 mg/L, the oxidation rate of Fe^2+^ in each reactor decreased.

### 3.6. The Influence of Fe^2+^ Oxidation on the Subsequent Neutralization Treatment

#### 3.6.1. The Influence of Fe^2+^ Oxidation on the Removal of Fe by Lime (CaO) Neutralization

Figure 8a shows the effect of CaO neutralization treatment on the simulated acid mine drainage before oxidation. The pH of the simulated acid mine drainage before oxidation was 1.96 and the Fe^2+^ content was 555 mg/L. During the neutralization process, we added CaO to 150 mL of the solution. When the added amount was 0.25 g, the pH of the solution rose to 6.40, and the Fe^2+^ in the solution decreased to 372 mg/L. When the added amount reached 0.3 g, the pH of the solution rose to 7.93 and the Fe^2+^ content dropped to 39 mg/L. When the added amount reached 0.35 g, the pH of the solution rose to 8.74, at which point the Fe^2+^ can all produce Fe(OH)_2_ precipitation.

Figure 8b shows the effect of the simulated acid mine drainage after oxidation by the CaO neutralization method. The oxidation of Fe^2+^ is an acid-consuming reaction. After 555 mg/L Fe^2+^ was completely oxidized, the pH of the solution rose from 1.96 to 2.01. When the amount of CaO added was 0.25 g, the pH of the solution rose to 4.06 and the Fe^3+^ decreased to 0. At this time, the solution did not contain soluble Fe. In Figure 8a, in the solution, only 1/3 of the Fe^2+^ is now precipitated.

Therefore, after the Fe^2+^ is oxidized to Fe^3+^, a good Fe removal effect can be achieved under low pH conditions, which in turn can reduce the amount of neutralizer used and reduce the burden of the subsequent sediment treatment.

#### 3.6.2. The Influence of Fe^2+^ Oxidation on the Removal of Fe by Limestone (CaCO_3_) Neutralization

Figure 9a shows the effect of the CaCO_3_ neutralization method on the simulated acid mine drainage before oxidation. When 0.4 g CaCO_3_ was added to 150 mL of unoxidized simulated acid mine drainage, the pH of the solution quickly increased from 1.96 to 4.99, and the Fe^2+^ had no obvious removal effect.

As shown in Figure 9a, the pH of the solution rises very slowly after rising to 4.99. Even when the amount of CaCO_3_ added reaches 2 g, the pH of the solution only rises to 5.40, and the dissolution rate of CaCO_3_ slows down with the increase in the pH.

From this experiment, it can be seen that there is no obvious precipitation in the solution, so the concentration of Fe^2+^ does not change significantly. Therefore, it can be seen that the limestone neutralization method is not effective in removing Fe^2+^.

Figure 9b shows the effect of simulated acid mine drainage after oxidation by the CaCO_3_ neutralization method. When the Fe^2+^ was oxidized to Fe^3+^, 0.4 g of CaCO_3_ was also added to 150 mL of solution, but the content of the soluble Fe (Fe^3+^) in the reaction system was reduced to 25 mg/L, and the removal efficiency reached 96%. When the added amount of CaCO_3_ reached 0.5 g, all the Fe^3+^ in the solution was precipitated.

#### 3.6.3. Advantage Analysis for the Enhanced Microbial Oxidation–Neutralization Method

Table 2 shows the difference in the precipitates obtained by simulating acid mine drainage before and after oxidation with different neutralizers (CaO, CaCO_3_). When the oxidized simulated acid mine drainage was treated with CaCO_3_, the resulting precipitated Fe(OH)_3_ was bright yellow, with a compact structure, a solid content of 5.50%, an SV_30_ of only 4%, and a CST of 8.9 s.

When CaO was used as a neutralizer, when the simulated acid mine drainage before oxidation was treated, the precipitate formed was Fe(OH)_2_, which is blackish green with a loose structure, a solid content of 2.75%, an SV_30_ of 24%, and a CST of 13.7 s. When the simulated acid mine drainage was processed after oxidation, the precipitate formed was Fe(OH)_3_, which is orange yellow in color, with a solid content of 1.60%, an SV_30_ of 28%, and a CST of 21.1 s.

It can be seen that, for Fe^2+^-rich acid mine drainage, *A. ferrooxidans* is used to oxidize Fe^2+^ to Fe^3+^, and then CaCO_3_ is used to directly neutralize it under low pH conditions. Not only is the price of the neutralizer low, but the generated precipitate also has a small volume (the volume is 1/6 to 1/7 when the CaO is directly used for neutralization), the structure of the precipitate is dense, the solid content is high, and the sedimentation performance and dehydration performance are good, so that the subsequent treatment and disposal costs are low. When CaO is used as a neutralizer, the colloidal precipitation structure is loose, with a high water content, which is difficult to compress and dehydrate, and is also difficult to handle.

## 4. Conclusions

When the influent Fe^2+^ concentration was about 900 mg/L and the temperature was 18 °C, under the premise of ensuring the Fe^2+^ oxidation efficiency, the maximum oxidation rate for Fe^2+^ in the bio-ceramsite, bio-volcanic stone, and bio-activated carbon packed columns are 301 mg/(L·h), 234 mg/(L·h), and 139 mg/(L·h). When the concentration of Fe^2+^ in the influent water dropped from 900 mg/L to about 600 mg/L and 300 mg/L, the oxidation rate of Fe^2+^ in each reactor decreased. Before the oxidation of Fe^2+^, the CaO, rather than CaCO_3_, is suitable for use as a neutralizer. When 0.25 g of CaO was added to 150 mL of simulated acid mine drainage before *A. ferrooxidans* oxidation, only 1/3 of the Fe^2+^ in the solution was precipitated; when 0.25 g of CaO was added to the drainage after oxidation, no soluble Fe existed in the solution. In contrast, after Fe^2+^ oxidation, CaCO_3_ can also be used as a neutralizing agent. When 0.5 g of CaCO_3_ was added to the acid mine drainage, all Fe^3+^ in the solution precipitated.

Compared with the traditional CaO direct neutralization method, the enhanced microbial oxidation-CaCO_3_ neutralization is an efficient technique for the acid mine drainage treatment. As there are differences in the sedimentation performance between Fe^2+^ and Fe^3+^, abundant *A. ferrooxidans*, are applied to oxidize Fe^2+^ to Fe^3+^ in the wastewater, with cheaper CaCO_3_ instead of CaO being used as a neutralizer. The resulting precipitate with a dense structure, high solid content, and good sedimentation performance is easily dehydrated and beneficial to the subsequent treatment and disposal.

## Figures and Tables

**Figure 1 ijerph-19-06543-f001:**
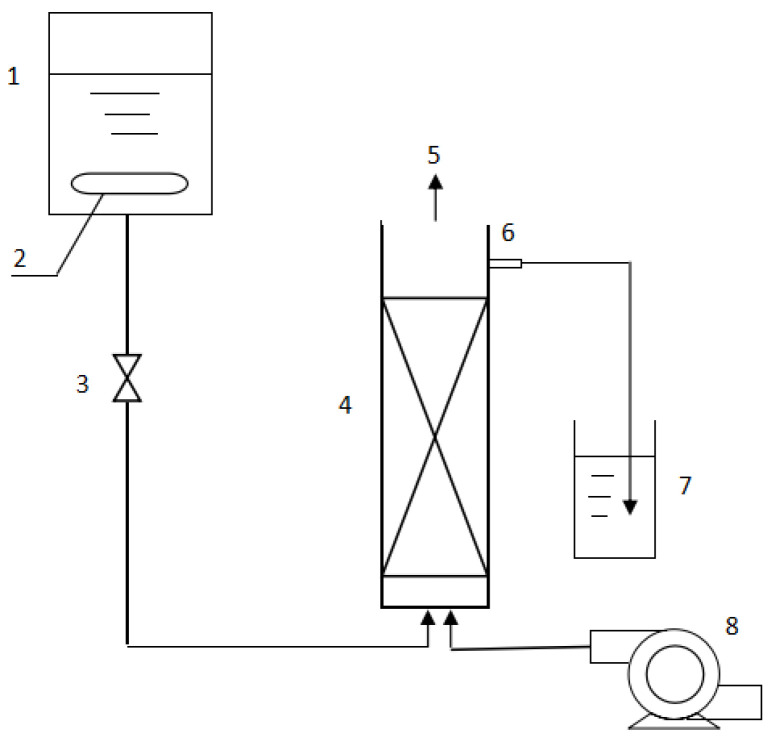
Schematic diagram of the packed-bed bioreactor. Note: 1. High slot. 2. Automatic constant temperature heating rod. 3. Valve. 4. Packed bed. 5. Air outlet. 6. Water outlet. 7. Low slot. 8. Air compressor.

**Figure 2 ijerph-19-06543-f002:**
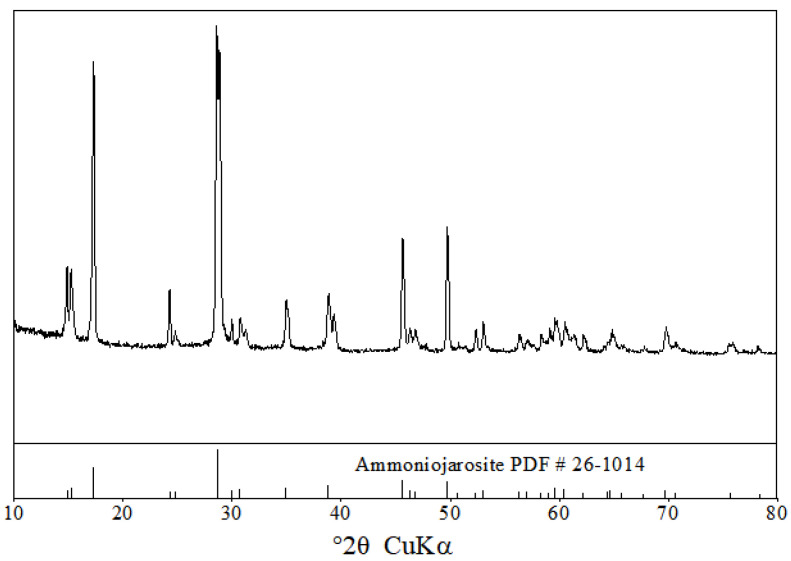
XRD patterns of precipitates.

**Figure 3 ijerph-19-06543-f003:**
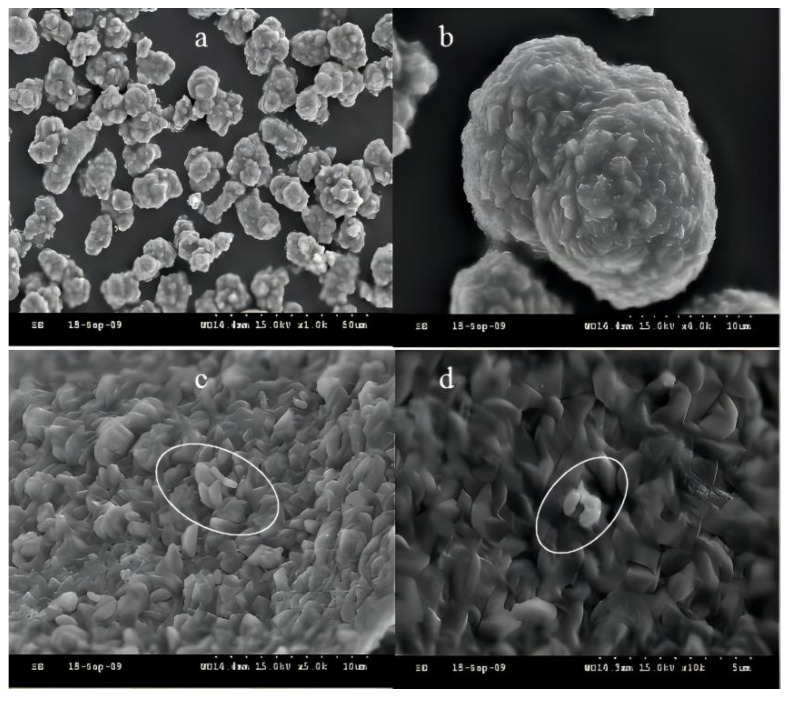
Scanning electron micrograph of jarosite formed in the presence of *A. ferrooxidans* ((**a**) ×1000; (**b**) ×4000; (**c**) ×5000; and (**d**) ×10,000).

**Figure 4 ijerph-19-06543-f004:**
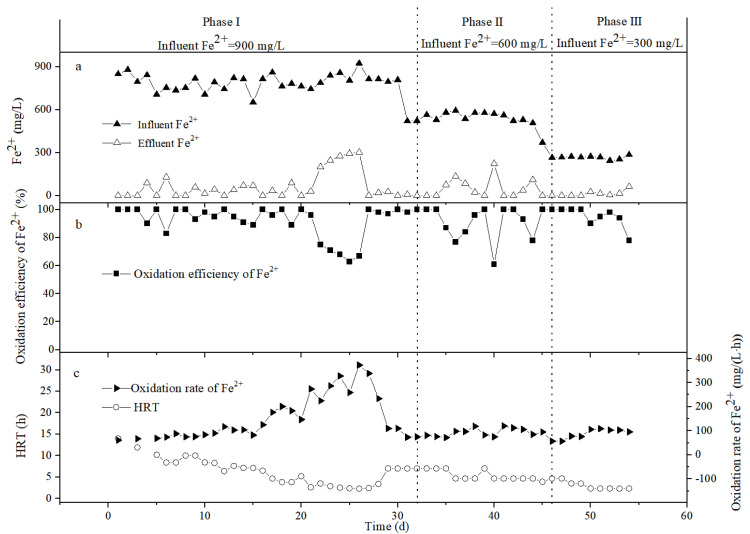
Variation of the Fe^2+^ concentration (**a**), Fe^2+^ oxidation efficiency (**b**), Fe^2+^ oxidation rate, and HRT (**c**) in the influent and effluent of the bio-volcanic rock packed column.

**Figure 5 ijerph-19-06543-f005:**
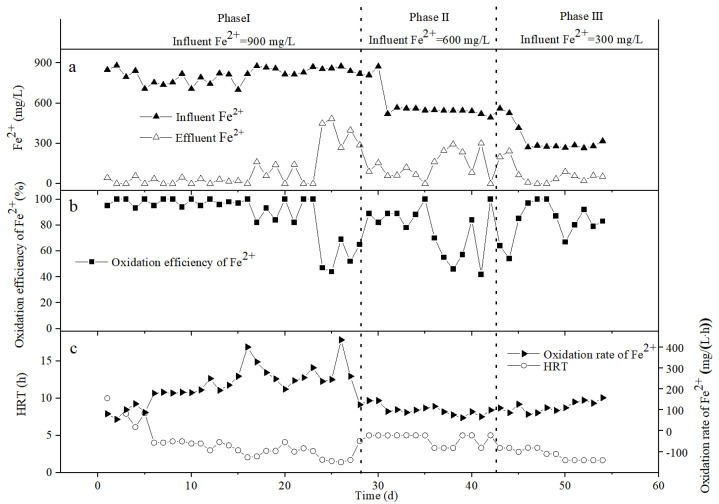
Variation of the Fe^2+^ concentration (**a**), Fe^2+^ oxidation efficiency (**b**), Fe^2+^ oxidation rate, and HRT (**c**) in the influent and effluent of the bio-ceramsite packed column.

**Figure 6 ijerph-19-06543-f006:**
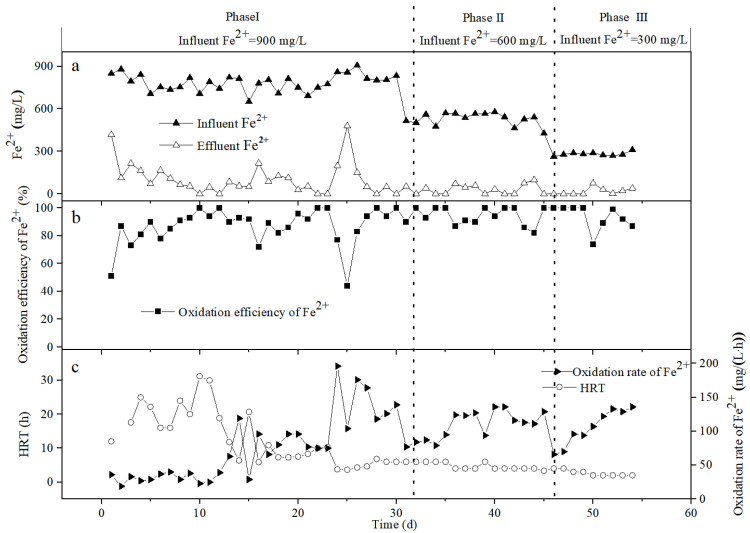
Variation of the Fe^2+^ concentration (**a**), Fe^2+^ oxidation efficiency (**b**), Fe^2+^ oxidation rate, and HRT (**c**) in the influent and effluent of the bio-activated carbon packed column.

**Figure 7 ijerph-19-06543-f007:**
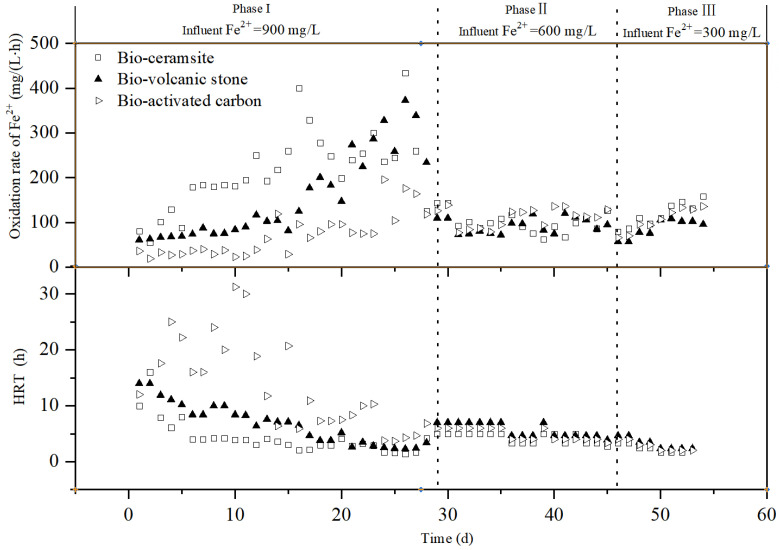
Variation in the oxidation rate of Fe^2+^ in each packed column under different influent concentrations and HRT conditions.

**Figure 8 ijerph-19-06543-f008:**
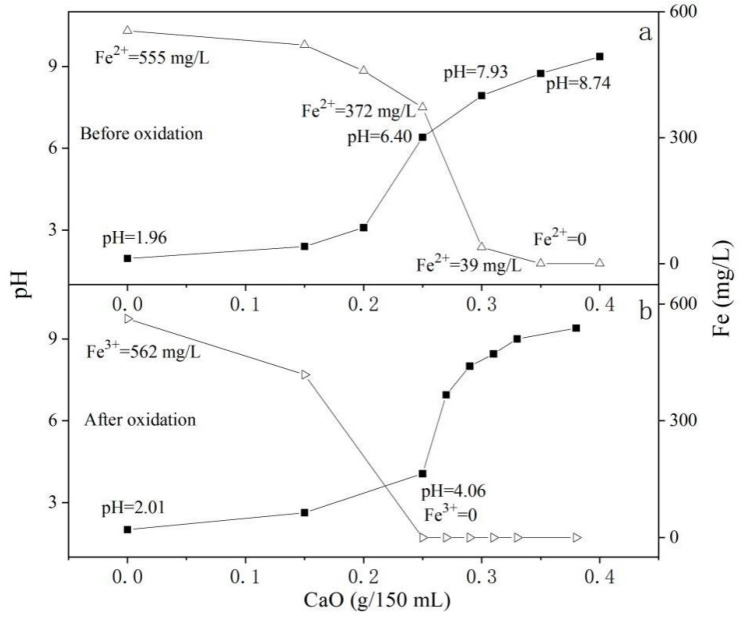
The effect of CaO amount on the pH and soluble Fe of the Fe^2+^-rich simulated acid mine drainage before (**a**) and after (**b**) oxidation.

**Figure 9 ijerph-19-06543-f009:**
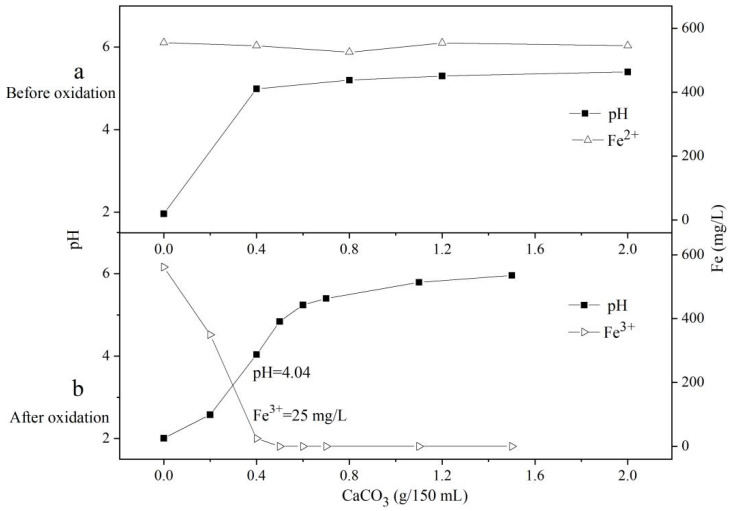
The effect of the CaCO_3_ amount on the pH and soluble Fe of the Fe^2+^-rich simulated acid mine drainage before (**a**) and after (**b**) oxidation.

**Table 1 ijerph-19-06543-t001:** Comparison of the oxidative capacity of the three packed columns.

Packed Column	Time (d)	HRT (h)	Influent Fe^2+^ (mg/L)	Effluent Fe^2+^ (mg/L)	Fe^2+^ Oxidation Efficiency	Fe^2+^ Oxidation Rate (mg/(L·h))
Bio-ceramsite	23	2.89	871	0	100%	301
Bio-volcanic stone	28	3.40	814	20	98%	234
Bio-activated carbon	30	6.00	834	0	100%	139

**Table 2 ijerph-19-06543-t002:** Analysis of the difference of the precipitates obtained under different neutralizers.

Neutralizer	The Precipitate Obtained by Neutralization after Oxidation (Containing Fe(OH)_3_)				The Precipitate Obtained by Neutralization before Oxidation (Containing Fe(OH)_2_)			
	Form	Solid Content	CST (s)	SV_30_	Form	Solid Content	CST(s)	SV_30_
CaCO_3_	Bright yellow with compact structure	5.50%	8.9	4%	/	/	/	/
CaO	Orange yellow With loose structure	1.60%	21.1	28%	Blackish green with loose structure	2.75%	13.7	24%

Note: No precipitate was formed when CaCO_3_ treated the simulated acid mine drainage before oxidation.

## Data Availability

The data presented in this study are available on request from the corresponding author.

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
