# Peer review of "Enhanced Microbial Oxidation–Neutralization Treatment of Acid Mine Drainage Rich in Ferrous Ions (Fe2+)"

_ijerph, 2022, doi:10.3390/ijerph19116543_

Round 1

Reviewer 1 Report

  1. This manuscript needs extensive style and English grammar elaboration to fix numerous grammar mistakes and typos.

-line 79-81: the meaning of sentence is unclear.

Line 160: speed of.

  1. Lines 88-96: why inlet Fe2+ concentration was decreasing with the increase in operating time? Needs to be elaborated.
  2. SEM images shown in Fig.3are not conclusive. SEM itself can show the particle’s morphology and and size but it cannot tell what it is made of. To make a statement about crystal composition, XRD or at least XPS should be performed.
  3. Fig.2c – I do not see how the authors determined that the particles are covered with bacteria. Again, staining with dye and confocal microscopy may be a better indication.
  4. lines 167-168: “the oxidation process of Fe2+ is also the process of growth and reproduction of A.ferrooxidans”- this phrase is not clear to me. The oxidation process cannot be the growth process.
  5. Fig 3c: why oxidation rate increased so high within the first 30 days of process?
  6. Fig 4b: how the oxidation rate units could be in expressed in %??? Could the authors explicitly describe the terms of oxidation rate and oxidation speed and explain the difference between them?
  7. Lines 249: “Since the Fe2+ concentration in the effluent water needs to be considered in practical applications, all the points where the Fe2+ oxidation rate was lower than 90% in the different reaction columns were deleted in Figure. 6, in order to compare the Fe2+ oxidation capacity when the device was operating in good conditions.” – I am not convinced that results could be just removed to fit a certain trend. A better justification is needed to explain this.
  8. Lines 296 –onwards. The authors are talking about the oxidation speed but in fact, Fig 6 shows oxidation rate. Are they the same? Please see my previous comment and explicitly elaborate this issue through the manuscruipt.
  9. Line 373 – sentence is not clear.

Reviewer 2 Report

Some Notes on the Manuscript. Check the text for possible errors / typos, e.g. 157 - "Fe oxidation rate" is given twice, instead of "Fe oxidation speed". Supplement the methodology with information on the device on which the SEM analysis was performed and the type of paper for CSK analysis - "special filter paper" is not enough. Was it Whatman blotting paper, e.g. grade 17 CHR, used in this type of analysis? I also suggest that in any further research, the influence of temperature in the range of 10-35 degrees C should be determined, but only for the most optimal conditions used in the presented article. 

Reviewer 3 Report

“Enhanced microbial oxidation-neutralization treatment of acid mine drainage rich in ferrous ions (Fe2+)” by He et al. is an interesting manuscript, but it contains some things that need to be discussed.

  • The quality of the English language needs to be improved.
  • There are many typos. These typos mostly refer to omitting a space between words or to adding spaces where they do not belong. For example:
    • The space is missing in:
      • Lines 3, 33, 36, 37, 38, 39, etc. Between word and parenthesis.
      • Lines 9, 11, 55, 78, 118, etc. Between “A.” and “ferrooxidans
      • Lines 36, 41, 43, 50, 52, 54, etc. Before the new sentence begins.
    • Spaces must be removed in:
      • Line 19. Before the comma.
      • Line 52. Before the period.
      • References 10, 11, 13, 14, 15, etc. Before “, et al.”
      • References 2, 4. After “et al”.
    • Line 38. Perhaps insert “had” instead of “has”?
    • Line 45. “Treatments” instead of “treatment”?
    • Line 60. Perhaps write “of the bacteria” instead of “of bacteria”?
    • Line 103. “Figure” instead of “Figureure”.
    • Line 170, 183, 232, 281, etc. There should be no period after the word “Figure”.
    • Line 178, 182, 187, etc. Please use subscripts for ordinal numbers (e.g., 22nd instead of 22nd).
    • Line 189. Please, replace “234 mg. /(L·h)” with “234 mg/(L·h)”.
    • Line 305, 313, 317, 327, 331, and 337. Replace “7(a)”, “7(b)”, “8(a)”, and “8(b)” with “7a”, “7b”, “8a”, and “8b”, respectively.
    • Reference 6. Use italics for “Chlorella vulgaris”.
  • Please replace “AMD” with “acid mine drainage”.
  • Line 15. “With the decrease of the Fe2+ concentration in the influent, the oxidation speed of Fe2+ in each reactor also decreased.” This is to be expected. Therefore, this statement should be omitted from the Abstract.
  • Lines 38 and 39. Please, use a period instead of a semicolon.
  • Line 41. What is “wetlan”?
  • Line 94. The authors should consider replacing “three stages” with “three phases”, since the stages are actually called PHASE I, PHASE II, and PHASE III.
  • Lines 101, 128, and 136. The authors used TFe, although this abbreviation was not introduced earlier in the manuscript.
  • Line 108. Please replace “The packed bed in the packed-bed bioreactor” with “The packed bed in the bioreactor”.
  • Line 151. Put this sentence in context. “ ferrooxidans was attached ( Figure 2c), and A. ferrooxidans just finished dividing ( Figure 2d).”
  • In most cases, authors have reported HRT values using two decimal places (for example 2.05 h). They should do the same for all HRT values.
  • Line 157. The authors wrote “Fe2+ oxidation time” twice. I believe that the second time should be replaced by “Fe2+ oxidation speed”.
  • Phrases like “in a period of days 1-20” sound strange.
  • Lines 252 and 291. Replace “ceramsite” with “bio-ceramsite”.
  • Lines 253 and 291. Replace “volcanic” with “bio-volcanic”.
  • Line 291. Replace “activated” with “bio-activated”.
  • Some sentences are written oddly:
    • Line 43. “However, among the above methods and materials, most of them only have a good treatment effect on AMD in experiments, and are limited to various factors and cannot be used in a wide range of engineering treatment, especially for ferrous ion-rich AMD processing remains a big challenge.”
    • Line 61. “The oxidized effluent was neutralized to determine the amount of neutralizing agent (lime or limestone) needed to neutralize the unit volume of water, the pH of the influent and effluent, the soluble Fe content, and the sedimentation and dehydration properties of the resulting sludge.”
    • Line 69. “For the analog AMD reference 9K medium preparation, i.e., (NH4)2SO40 g/L, KCl 69 0.10 g/L, K2HPO4 0.50 g/L, Ca(NO3)2·4H2O 0.01 g/L, MgSO4·7H2O 0.50 g/L, the Fe2+ concentration (added in the form of FeSO4·7H2O) was set to 900 mg/L, 600 mg/L, and 300 mg/L, according to the application, and the initial pH was adjusted to 1.8 with H2SO4.”
    • Line 79. “After aerated culture until the Fe2+ was completely oxidized,…”
    • Line 84. “When the rate does not increase significantly, it can be regarded as successfully fixed and can be transferred to the packed-bed bioreactor.”
    • Line 253. “…activated carbon is particles with a particle size of about 0.1 cm.”
  • If I understood correctlyl, one of the results of the experiments was pH values at which Fe(OH)2 and Fe(OH)3 completely precipitated. If so, I would like the authors to explain why it was necessary to perform the experiment, since these values can be calculated from the values of solubility products.
  • The references are not written according to the instructions. This refers to the names of the authors cited.

The reviewer suggests major revision of the manuscript.

Round 2

Reviewer 1 Report

I believe that after authors' revisions this manuscript can by accepted in its present form.

Reviewer 3 Report

The authors improved the manuscript significantly.

The manuscript can be accepted for publication.